# NON-DEEP VISION TRANSFORMERS

## ABSTRACT

Vision Transformers (ViTs) achieve remarkable performance on image classification tasks but suffer from computational inefficiency due to their deep architectures. While existing approaches focus on token reduction or attention optimization, the fundamental challenge of reducing architectural depth while maintaining representation capacity remains largely unaddressed. We propose a novel structural reparameterization approach that enables training of parallel-branch transformer architectures which can be collapsed into efficient single-branch networks during inference. Our method progressively joins parallel branches at the inputs of non-linear functions during training. This allows the reparameterization of both the multi-head self-attention (MHSA) and feed forward network (FFN) modules during inference without approximation loss. When applied to DeiT-Tiny, our approach compresses the model from 12 layers to as few as {3, 4, 6} while preserving accuracy, delivering up to 37% lower inference latency on mobile CPUs for ImageNet-1K classification. Our findings challenge the conventional wisdom that transformer depth is essential for strong performance, opening new directions for efficient ViT design.

## 1 INTRODUCTION

Vision Transformers (ViTs) have transformed computer vision by modeling long-range dependencies and achieving state-of-the-art performance across diverse tasks (Dosovitskiy et al., 2020). However, their high computational cost, driven by quadratic self-attention and deep sequential architectures, poses challenges for deployment on resource-constrained devices. This issue is especially critical for edge and mobile deployment, where devices such as smartphones, embedded processors, and ARM-based systems must operate under strict compute, memory, and latency budgets for real-time applications. Even the most compact ViTs remain difficult to deploy efficiently on these platforms. For example, TinyViT (Wu et al., 2022), one of the shallowest ImageNet-1K ViTs, still has 12 layers and 5.7M parameters. While pruning and quantization reduce storage and FLOPs, such models often struggle to meet real-time constraints or run efficiently on ARM processors and mobile CPUs (Nandakumar et al., 2019; Lyu et al., 2022; Wu et al., 2022). This depth–latency bottleneck calls for a paradigm shift toward ultra-shallow architectures that reduce sequential operations while retaining competitive accuracy.

Beyond traditional mobile deployment, non-deep networks are increasingly important for emerging paradigms like in-memory computing and photonic computing (Negi et al., 2024; Khilo et al., 2012). These analog systems face high ADC/DAC overhead, with ADCs consuming up to 60% of energy and 80% of area in compute-in-memory accelerators (Saxena et al., 2021). Noise also accumulates with depth, degrading accuracy in deeper networks (Agrawal & Roy, 2019). Similarly, photonic systems require costly optical-to-electrical conversions at each layer, which compounds with depth (Shen et al., 2017). For such platforms, ultra-shallow networks (4–6 layers) strike a balance between efficiency and computational capability. While recent works focus on optimizing attention mechanisms (Wang et al., 2021) or reducing tokens (Rao et al., 2021), they largely assume fixed deep architectures. Traditional ViT efficiency efforts—architectural tweaks, training strategies, and post-training compression like pruning or quantization—optimize within this deep paradigm rather than questioning its necessity. Recent studies show that parallel architectures can match the performance of deeper models while reducing sequential computation (Anwar et al., 2021). Meanwhile, structural reparameterization allows complex training-time networks to collapse into efficient inference-time models (Ding et al., 2021b). These advances suggest an opportunity to rethink ViT design by combining parallel training with deployment-oriented reparameterization.

**To the best of our knowledge, this work represents the first attempt to demonstrate competitive ViT performance with 6 layers or fewer.** Extending existing efficiency approaches to such extreme depth reduction is non-trivial and often counterproductive. Knowledge distillation methods, while effective for moderate compression (Touvron et al., 2021; Hao et al., 2022), face severe limitations when applied to ultra-shallow students. Recent distillation work shows that DeiT-Tiny students with 12 layers can achieve at most 76.5% ImageNet accuracy even with sophisticated manifold distillation (Hao et al., 2022), and performance degrades rapidly as student depth decreases below 8-10 layers (Chen et al., 2022a). Traditional pruning approaches (Liu et al., 2022; 2024) are designed for moderate depth reduction (removing 20-30% of layers) and cannot handle the extreme compression ratios required for 6-layer models without catastrophic accuracy loss. Other depth compression approaches, such as network overparameterization Yu et al. (2025) and non-linearity manipulation Fu et al. (2022); Bhardwaj et al. (2022), have yet to demonstrate scalability to ViTs. Structural modifications like depth-wise convolutions (Mahmood et al., 2024) and hybrid CNN-ViT architectures require fundamental changes to the transformer paradigm, making them incompatible with standard ViT deployments.

**Our Contributions.** We propose a novel framework that leverages *progressive structural reparameterization* to enable ultra-efficient Vision Transformers (ViTs). Our approach trains parallel-branch transformer blocks that gradually merge during training and ultimately collapse into standard single-branch architectures for lightweight inference. The core idea is to progressively join these branches at the inputs of key non-linearities—softmax in attention mechanisms and GeLU in feedforward layers—ensuring exact algebraic reparameterization with no approximation loss. This allows branches to remain diverse during early training, promoting richer feature learning, while gradually unifying their representations as training proceeds. Using this framework, we show that *non-deep ViTs* with just 4–6 layers can match the accuracy of 12-layer DeiT-Tiny models on ImageNet-1K while delivering substantial deployment benefits. Specifically, at iso-accuracy, our reparameterized models reduce end-to-end inference latency by 39% on ARM processors and 37% on mobile CPUs, and improve throughput by 64% compared to the original ViT. This challenges the long-standing assumption that depth is essential for transformer performance and opens up new opportunities for deployment in mobile and edge computing, as well as emerging analog and photonic accelerators, where strict memory, latency, and energy constraints dominate.

## 2 RELATED WORK

**ViT Efficiency:** Research on efficient ViTs has explored multiple levels of optimization. *Token-level methods*, such as DynamicViT (Rao et al., 2021) and TokenLearner (Ryoo et al., 2021), reduce the number of tokens processed through dynamic selection or learned aggregation. *Attention-level approaches* address the quadratic cost of self-attention using linear variants (Katharopoulos et al., 2020), sparse patterns (Child et al., 2019), or multi-scale designs (Wang et al., 2021). Recent architectures challenge core transformer assumptions: EfficientFormer (Lyu et al., 2022) achieves mobile-speed inference with dimension-consistent design, SwiftFormer (Shaker et al., 2023) replaces key-value matrix multiplication with element-wise operations for linear complexity, SHViT (Lee et al., 2024) uses single-head attention for speed, and MoR-ViT (Zhang et al., 2024) applies mixture-of-recursions for dynamic depth allocation. Hybrid CNN-transformer models combine local convolutional inductive biases with global modeling. Examples include CvT (Wu et al., 2021), CMT (Guo et al., 2022), MobileViT (Mehta & Rastegari, 2022), and NextViT (Li et al., 2022), which strategically merge CNN and transformer blocks for better accuracy-efficiency trade-offs. Another major direction for efficiency is pruning and slimming of ViTs, which focuses on identifying redundant components to reduce FLOPs and parameters. Self-Slimmed ViT (Zong et al., 2022) prunes channels and attention heads using sparsity-inducing gates, while Unified Visual Transformer Compression (Yu et al., 2022) jointly optimizes pruning, quantization, and knowledge distillation in a single framework. X-Pruner (Yu & Xiang, 2023) introduces a differentiable search process to prune layers, heads, and tokens simultaneously, and LPViT (Xu et al., 2024) leverages frequency-domain low-pass filtering to reduce computational cost. While these methods successfully lower *within-layer* computation or partially remove layers, they rely on approximations and post-hoc distillation, and are fundamentally constrained by the assumption of deep sequential models. In contrast, our approach reparameterizes *entire branches* during training and produces an exactly equivalent shallow transformer at inference, enabling aggressive depth reduction (down to 3–6 layers) while retaining compatibility with standard ViT inference pipelines.

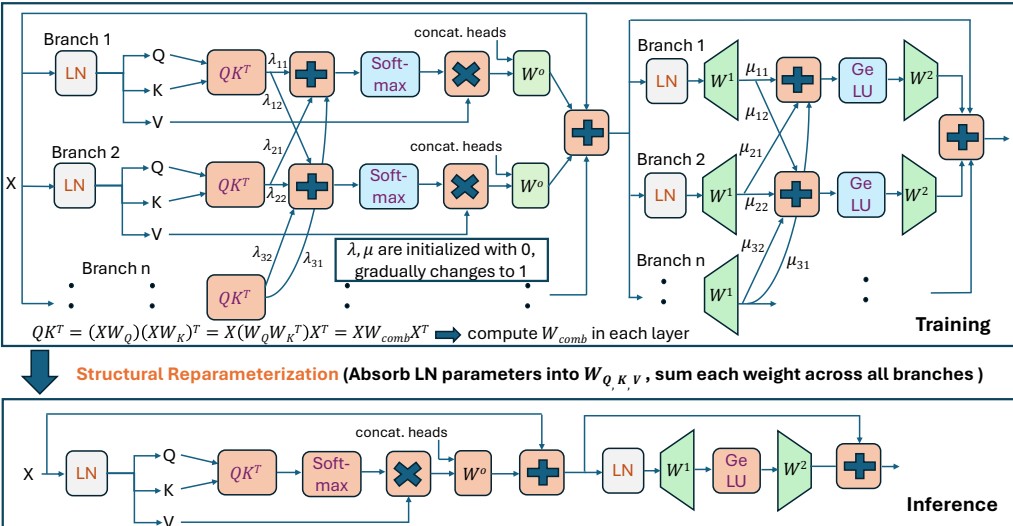

Figure 1: Proposed Depth Compression Framework for ViTs

**Structural Reparameterization:** Structural reparameterization decouples training-time architectural complexity from inference-time efficiency. RepVGG (Ding et al., 2021b) pioneered this idea by showing that multi-branch training structures can be algebraically collapsed into single-branch inference networks without performance loss. Diverse Branch Block (DBB) (Ding et al., 2021a) extended this with multi-scale transformations during training, while OREPA (Hu et al., 2023) introduced online reparameterization to reduce training-time memory costs. UniRepLKNet (Ding et al., 2024) showed that extremely large kernels can be efficiently compressed for deployment, and RepViT (Wang et al., 2024) adapted these ideas to mobile backbones such as MobileNetV3. Extensions to transformers have been limited: FastViT (Vasu et al., 2023) introduced RepMixer for token mixing, while RePaViT (Chang et al., 2024) applied reparameterization only to FFN layers.

**Non-Deep Network Architectures:** A growing body of work challenges the long-standing assumption that increasing network depth is the only path to strong performance. Early studies such as WideResNets (Zagoruyko & Komodakis, 2016) highlighted the fundamental trade-off between width and depth in convolutional networks, demonstrating that sufficiently wide but shallow architectures can match or even surpass the accuracy of deep, narrow counterparts. Building on this idea, ParNet (Anwar & Hwang, 2021) showed that parallel subnetworks can achieve competitive accuracy with drastically reduced sequential depth by exploiting modern hardware for parallel computation. More recently, Wightman et al. (Wightman et al., 2022) demonstrated that carefully optimized training strategies allow even relatively shallow ResNets to rival deep models, although they still require more than 12 layers to remain competitive. Certain pruning and slimming approaches, such as X-Pruner (Yu & Xiang, 2023), implicitly take advantage of this width–depth trade-off by adaptively thinning or widening layers to balance efficiency and capacity. However, these methods stop short of explicitly constructing *ultra-shallow* architectures and instead focus on gradual compression of deep models. From a theoretical standpoint, recent results on shallow Vision Transformers (Li et al., 2023) formalize the conditions under which depth becomes less critical. Specifically, when (i) patch embeddings are sufficiently expressive and (ii) attention and MLP widths scale appropriately with token count, shallow transformers can achieve competitive sample complexity, with depth contributing only sublinearly once token- and width-dependent margins dominate. Our design aligns with these theoretical insights: the final deployed model uses only $L/n$ sequential layers while preserving the width and tokenization of the original $L$-layer ViT, thus satisfying the scaling preconditions for shallow generalization and efficient deployment.

## 3 PROPOSED METHOD

### 3.1 OVERVIEW AND CHALLENGES

ViTs achieve strong performance through deep stacks of transformer blocks, each containing multi-head self-attention (MHSA) and feed-forward network (FFN) modules. However, this depth creates severe computational bottlenecks for deployment. Our approach mitigates this by training parallel transformer branches that are later collapsed into a single branch at inference, drastically reducing depth while preserving representational capacity. The key idea is that, with proper constraints, parallel branches can be *algebraically reparameterized* into one equivalent branch without approximation

error. Consider two parallel transformer blocks processing the same input $X$. Each branch has independent parameters and computes its own transformations, but we seek to merge them into a single set of operations. If the blocks contained only linear operations, this would be trivial—we could simply sum the weight matrices. In practice, three challenges prevent naive reparameterization: *First, non-linearities*—softmax in MHSA and GELU in FFN—break the linearity needed for direct combination. For example, $\texttt{softmax}(QK_A^T) + \texttt{softmax}(QK_B^T) \neq \texttt{softmax}(QK_A^T + QK_B^T)$. *Second, Layer Normalization (LN)* within each block introduces mismatched normalization statistics across branches, disrupting fusion. *Third, multi-head attention concatenates head outputs*, making it impossible to directly align and merge branches since $[H_{A,1}; \ldots; H_{A,h}]$ from Branch A cannot be cleanly combined with Branch B.

We address these challenges through a unified framework that combines architectural modifications with progressive training. LN operations are extracted from parallel branches, and MHSA is reformulated using blockwise operations to replace concatenation with summation. During training, we progressively join branch outputs at the inputs to non-linear functions, so that by the end of training, all branches receive identical inputs. This guarantees exact algebraic reparameterization while leveraging the diversity and capacity benefits of parallel training.

### 3.2 ARCHITECTURAL MODIFICATIONS FOR REPARAMETERIZATION

**Layer Normalization Factorization**: To enable structural reparameterization, we must first address the fundamental incompatibility created by LN operations within parallel branches. The standard transformer architecture applies LN before each sub-module: one before MHSA and one before FFN. In a parallel-branch setting, if these normalizations occur within branches, each branch computes different normalization statistics, preventing algebraic combination. Our solution restructures the architecture by extracting LN operations from within the parallel branches and positioning them before the branching points. This ensures both branches receive identically normalized inputs. To maintain mathematical equivalence while enabling this restructuring, we absorb the LN parameters into adjacent linear transformations.

*For FFN modules*, we absorb the LN parameters forward into the FFN's first linear transformation. This is possible because the FFN begins with a linear layer that can directly incorporate the normalization's scale and bias parameters. The original computation $W_1 \cdot \text{LN}(x)$ becomes:

$$W_1^{\text{new}} = W_1 \cdot \text{diag}(\gamma), \quad b_1^{\text{new}} = W_1\beta + b_1 \tag{1}$$

where $\text{diag}(\gamma)$ creates a diagonal matrix from the scale parameters. This absorption maintains the mathematical equivalence while eliminating LN from within the parallel branches. *For MHSA modules*, however, forward absorption is not feasible because it would disrupt the specific mathematical structure required for attention reparameterization. The attention mechanism relies on the $XWX^T$ formulation (detailed in Section 3.3), where inserting LN parameters would break this clean algebraic form needed for branch combination. Therefore, we absorb the MHSA LN parameters backward into the previous transformer block's FFN layer. Specifically, if the previous layer computes $y = W_2h + b_2$, and this is followed by LN with parameters $\gamma_{\text{mhsa}}, \beta_{\text{mhsa}}$ before the next MHSA module, we modify:

$$W_2^{\text{new}} = \text{diag}(\gamma_{\text{mhsa}}) \cdot W_2, \quad b_2^{\text{new}} = \gamma_{\text{mhsa}} \odot b_2 + \beta_{\text{mhsa}} \tag{2}$$

For the first MHSA module, where no previous FFN exists, we absorb these parameters into the patch embedding layer. This systematic absorption ensures that our restructured architecture computes exactly the same function as the original while enabling parallel branches to process identical normalized inputs, with each module's specific structural requirements properly addressed. Importantly, this approach completely eliminates the computational overhead of Layer Normalization operations while adding no additional computational cost to the linear/non-linear transformations.

**Blockwise Multi-Head Attention Reformulation**: The standard multi-head attention implementation creates a structural barrier to reparameterization through its use of concatenation operations. After computing attention for each head independently, the outputs are concatenated before the final projection, which prevents clean algebraic combination of parallel branches. Our solution reformulates multi-head attention using blockwise operations that eliminate concatenation entirely. Instead of concatenating head outputs before applying a monolithic output projection matrix $W^O \in \mathbb{R}^{d \times d}$, we partition this matrix into head-specific blocks: $W^O = [W_1^O, W_2^O, \ldots, W_h^O], \quad \text{where } W_i^O \in$

$\mathbb{R}^{d_h \times d}$. Each head's output is now independently projected and the final output is computed through summation rather than concatenation followed by projection:

$$\text{Output} = \sum_{i=1}^{h} H_i W_i^O = \sum_{i=1}^{h} \text{softmax}\left(\frac{Q_i K_i^T}{\sqrt{d_k}}\right) V_i W_i^O \tag{3}$$

This reformulation is mathematically equivalent to standard multi-head attention but replaces the non-linear concatenation with linear summation, enabling clean reparameterization after training. This blockwise computation introduces no additional computational overhead, as modern GPUs and edge accelerators can efficiently execute these head-wise operations in parallel through native block matrix multiplication primitives, eliminating any synchronization barriers that might arise from the summation operation.

### 3.3 PROGRESSIVE JOINING FRAMEWORK

With the architectural modifications in place, we now present our progressive joining mechanism that enables structural reparameterization while maintaining diverse representations during training. The key innovation lies in how we handle the computation before and after non-linear functions differently to enable algebraic combination.

**Reparameterization Before Softmax**: For the attention mechanism, the critical insight is to reorder the computation of attention scores to enable pre-softmax combination. In standard attention, each branch would compute: $(QK^T)_A = Q_A K_A^T = (XW_A^Q)(XW_A^K)^T$. We restructure this computation by recognizing that this can be rewritten as: $(QK^T)_A = XW_A^Q W_A^{K^T} X^T = XW_A X^T$, where $W_A = W_A^Q W_A^{K^T}$ is a combined weight matrix. This reordering is crucial because it transforms the attention score computation into a form $XWX^T$ that can be algebraically combined across branches. Since computing in the $XWX^T$ formulation operates in the full embedding dimension $d$ rather than the per-head dimension $d_h$, this approach is computationally more expensive than traditional $QK^T$ computation. For DeiT-Tiny Wu et al. (2022) with $d$=192, $n$=197 tokens, the $XWX^T$ approach requires 58.8M operations compared to 43.9M for traditional $QK^T$, representing a 33% increase. However, this allows offline precomputation and storage of the combined weight matrix $W$, reducing the dataflow from three matrix multiplications ($X{\rightarrow}Q$, $X{\rightarrow}K$, $Q{\times}K^T$) to two ($X{\rightarrow}XW$, $XWX^T$), potentially improving efficiency. Since FFN operations dominate the compute (116.1M operations vs 43.9M for attention in DeiT-Tiny), our MHSA overhead is negligible, incurring $\sim$7% of the total ViT computation. A similar proportion of overhead is also observed in terms of the number of parameters.

During training with progressive joining parameter $\lambda(t)$, both branches compute:

$$\text{Branch A and B attention scores:} \quad XW_A X^T, \ XW_B X^T \tag{4}$$

$$\text{Branch A and B softmax receives:} \quad X(W_A{+}\lambda(t)W_B)X^T, \ X(W_B{+}\lambda(t)W_A)X^T \tag{5}$$

One concern of the joining is scale amplification: merging multiple score distributions increases variance and can destabilize training. Assuming $W_A$ and $W_B$ are identically distributed, the effective variance of the combined attention logits grows with $(1 + \lambda^2(t))$. Thus, the rectified pre-softmax scale becomes $\sqrt{1 + \lambda^2(t)}\sqrt{d_k}$. $\lambda(t)$ follows a linear schedule: $\lambda(t) = t$, where $t$ is uniformly varied from 0 to 1 during training. This schedule ensures smooth transition from independent branch processing ($\lambda$=0) to fully joined computation ($\lambda = 1$), allowing branches to develop complementary features while gradually aligning their computations for perfect reparameterization. At convergence when $\lambda = 1$, both branches receive $X(W_A + W_B)X^T$, enabling perfect reparameterization with $W_{\text{combined}} = W_A + W_B = W_A^Q W_A^{K^T} + W_B^Q W_B^{K^T}$.

**Reparameterization After Softmax**: After the softmax operation, both branches produce identical attention weights when $\lambda = 1$. For each head $i$, the attention computation becomes:

$$H_{i,\text{combined}} = \text{softmax}\left(\frac{X(W_{A,i}^Q W_{A,i}^{K^T} + W_{B,i}^Q W_{B,i}^{K^T})X^T}{2\sqrt{d_k}}\right) \cdot X(W_{A,i}^V + W_{B,i}^V) \tag{6}$$

where $H_{i,\text{combined}}$ is the combined attention output for head $i$. Since both branches now compute identical softmax outputs for each head, the value projections can be directly combined as

$W_{i,\text{combined}}^V = W_{A,i}^V + W_{B,i}^V$. The final output across all heads, using our blockwise reformulation:

$$\text{Output}_{\text{combined}} = \sum_{i=1}^{h} H_{i,\text{combined}} \cdot W_{i,\text{combined}}^O = \sum_{i=1}^{h} H_{i,\text{combined}} \cdot (W_{A,i}^O + W_{B,i}^O) \tag{7}$$

where $W_{i,\text{combined}}^O = W_{A,i}^O + W_{B,i}^O$ is the combined output projection for head $i$. The blockwise structure ensures that each head's computation remains independent and can be algebraically combined after training.

**Feed-Forward Network Reparameterization**: For the feed-forward network, progressive joining operates similarly. Each branch computes its first-layer output independently, but these values are progressively combined before the GELU activation:

$$\text{Branch A and B computes:} \quad h_A = W_{1A}x, \ h_B = W_{1B}x \tag{8}$$
$$\text{Branch A and B GELU receives:} \quad h_A + \lambda(t) \cdot h_B, \ h_B + \lambda(t) \cdot h_A \tag{9}$$

At convergence when $\lambda=1$, both branches apply GELU to $h_A + h_B = (W_{1A} + W_{1B})x$, enabling reparameterization with $W_{1,\text{combined}} = W_{1A} + W_{1B}$. Since both branches produce identical GELU outputs, the second layer can be combined as $W_{2,\text{combined}} = W_{2A} + W_{2B}$. Please refer to Algorithm 1 in Appendix for additional details on our structural reparameterization.

### 3.4 IMPLEMENTATION DETAILS

Our framework transforms a standard $L$-layer transformer into a compact $L/n$-layer model for deployment, where $n$ is the compression factor. During training, the original $L$ sequential layers are replaced by $L/n$ parallel blocks, each with $n$ branches, maintaining **identical parameter count and FLOPs** to the original model. After training, we **reparameterize** by summing branch weights, yielding a standard $L/n$-layer sequential transformer compatible with existing inference frameworks. For example, compressing DeiT-Tiny's 12 layers to 6 layers uses 2-branch blocks, where each block has twice the parameters of a single layer, keeping the total count constant. The only added cost during training is temporary storage of activations for all branches, mitigated by having fewer blocks and efficient tensor-parallel execution.

We adopt progressive joining immediately after pre-training, with a 10k-step warmup, 50k-step adjustment phase. Each block processes input $X$ by computing all $n$ branch transformations in parallel, progressively combining outputs, and summing them to form the block's output. Gradients naturally flow through this mechanism, implicitly regularizing branches by encouraging complementary learning. Reparameterization is computationally trivial, requiring only a single weight summation per block: $W_{\text{combined}} = \sum_{i=1}^{n} W_i$. The final compressed model has significantly fewer sequential layers, reducing latency and memory while preserving accuracy. Our approach is compatible with standard optimizations such as mixed-precision, gradient accumulation, and distributed data parallelism. Progressive joining also acts as a regularizer, enabling the compressed model to match or even exceed the performance of the deeper baseline.

## 4 EXPERIMENTS

We conduct extensive experiments to validate the effectiveness of the proposed re-parameterized ViT across a diverse set of tasks, including supervised fine-tuning on ImageNet-1K Deng et al. (2009), self-supervised pre-training, transfer learning to several small-scale classification benchmarks, and dense prediction tasks such as object detection and instance segmentation. All training and fine-tuning experiments are performed on 8 NVIDIA H200 GPUs, each with 141 GB of memory. Detailed experimental configurations are provided in Appendix A. To assess latency and throughput, we benchmark our models on three representative processors spanning a wide computational spectrum: a GPU (RTX 4080), a CPU (Intel i9-9900X, single-thread), and an ARM processor (Cortex-A72, single-thread). Latency is reported for batch size 1, while throughput is measured using a batch size of 32.

**Model setup and notation**: We adopt DeiT Touvron et al. (2022) for supervised learning and MAE He et al. (2022) for self-supervised learning, following the efficient design principles of Wang et al. (2023). Even lightweight ViTs are configured with 12 attention heads, and during fine-tuning, global average pooled features replace the class token for improved performance. For MAE, we add knowledge distillation from an MAE-Base teacher by transferring attention maps, which boosts downstream performance over DeiT-based models.

In multi-branch settings, the distillation loss is computed as the mean squared error between the teacher's attention maps and the averaged maps across branches. We adopt a unified naming scheme for clarity. A model with $l$ layers, knowledge distillation, and our re-parameterization is written as `D-arch-`$l$`-R`. Depth-scaling baselines

Table 1: Evaluation of our re-parameterized ViTs on ImageNet-1K with 300-epoch fine-tuning. Our low-depth models deliver substantially higher throughput and lower latency compared to the 12-layer DeiT-Tiny baseline, while maintaining comparable or better accuracy.

| Methods | #parameters | FLOPs | Throughput on GPU (fps) | Latency on CPU (ms) | Latency on ARM (ms) | Accuracy Top-1 (%) |
|---|---|---|---|---|---|---|
| DeiT-Tiny (12 layers) | 5.7M | 1.08G | 1723 | 66.1 | 793 | 72.2 |
| DeiT-6 D-MAE-6 | 3.1M | 556M | 2932 | 41.3 | 456 | 63.4 68.6 |
| DeiT-4 D-MAE-4 | 2.2M | 381M | 3756 | 32.2 | 361 | 54.2 61.0 |
| DeiT-3 D-MAE-3 | 1.7M | 293M | 4114 | 28.4 | 289 | 38.7 53.8 |
| D-MAE-6-R D-MAE-4-R D-MAE-3-R | 3.3M 2.4M 1.9M | 595M 415M 322M | 2840 3592 3884 | 41.7 33.4 31.0 | 482 397 347 | **72.4** **67.0** **60.5** |

without re-parameterization are denoted as `D-arch-`$l$. Wider variants with $n$ parallel branches are represented as `D-arch-`$l$`-`$n$.

**Evaluation on ImageNet-1K**: Table 1 highlights the performance of our re-parameterized ViTs on the ImageNet-1K classification task, with a focus on latency and throughput across different hardware platforms. While the baseline DeiT-Tiny model with 12 layers achieves a top-1 accuracy of 72.2%, it suffers from significant latency (66.1 ms on CPU, 793 ms on ARM) and low GPU throughput (1723 fps), making it impractical for real-time or edge deployments. Our re-parameterized models demonstrate that substantial depth reduction can be achieved without sacrificing accuracy. Specifically, **D-MAE-6-R** matches and slightly outperforms DeiT-Tiny with a top-1 accuracy of **72.4%**, while reducing CPU latency by 37% (from 66.1 ms to 41.7 ms) and improving GPU throughput by 65% (from 1723 fps to 2840 fps). Similarly, ARM latency improves by 39% (from 793 ms to 482 ms), making our approach especially attractive for edge and mobile platforms. Even more aggressive depth reductions to 4 or 3 layers (D-MAE-4-R and D-MAE-3-R) maintain competi-

Table 2: Comparisons with previous SOTA networks with similar top-1 accuracy on ImageNet-1K. Our models achieve competitive accuracy while significantly reducing parameter count and FLOPs.

| Methods | #param. | FLOPs | Acc. Top-1 (%) |
|---|---|---|---|
| *Supervised & convolutional models* | | | |
| MobileNetV2 Sandler et al. (2018) | 3.5M | 310M | 72.0 |
| FasterNet-T0 Chen et al. (2023) | 3.9M | 340M | 71.9 |
| EtinyNet-M1 Xu et al. (2023) | 3.9M | 117M | 65.5 |
| MicroNet-M3 Li et al. (2021b) | 2.6M | 21M | 62.5 |
| *Supervised & convolutional ViT/pure ViT models* | | | |
| Mobile-Former-96M Chen et al. (2022b) | 4.6M | 96M | 72.8 |
| MobileViT-XS Mehta & Rastegari (2021) | 2.3M | 1.05G | 74.8 |
| PVTv2-B0v Wang et al. (2022) | 3.4M | 600M | 70.5 |
| EdgeViT-XXS Pan et al. (2022) | 4.1M | 600M | 74.4 |
| RePa-DeiT-Tiny/0.5 Xu et al. (2025) | 4.4M | 800M | 69.4 |
| DeiT-Tiny Touvron et al. (2021) | 5.7M | 1.08G | 72.2 |
| DeiT-6-R | 3.1M | 595M | 70.2 |
| DeiT-4-R | 2.2M | 415M | 64.1 |
| DeiT-3-R | 1.7M | 322M | 58.5 |
| *Self-supervised & pure ViT models* | | | |
| D-MAE-Tiny Wang et al. (2023) | 5.7M | 1.08G | 78.4 |
| D-MAE-6-R | 3.1M | 595M | 72.4 |
| D-MAE-4-R | 2.2M | 415M | 67.0 |
| D-MAE-3-R | 1.7M | 322M | 60.5 |

tive accuracy (**67.0%** and **60.5%**, respectively), while delivering progressively higher throughput and lower latency. This demonstrates the scalability of our approach, where reducing depth directly translates to deployment efficiency gains.

**Comparison with prior works**: We evaluate our approach on the ImageNet-1K classification task, where supervised models are directly re-parameterized after pretraining without additional fine-tuning, while self-supervised models are fine-tuned for 300 epochs post-reparameterization. Table 2 compares our low-depth, re-parameterized ViTs with previous state-of-the-art convolutional networks, hybrid convolutional-transformer models, and pure ViTs. Convolution-based models (e.g., MobileNetV2, FasterNet-T0) and hybrid CNN-ViT models (e.g., EdgeViT, MobileViT) tend to have lower FLOPs because convolution kernels are spatially shared through sliding-window operations. In contrast, pure ViTs rely on token-mixing via matrix multiplications or MLPs with no weight sharing, which inherently results in higher FLOPs for the same representational capacity. However, many convolutional and hybrid models achieve their efficiency through custom operators and irregular memory access patterns, which are often difficult to optimize for GPUs and edge devices. As a result, their theoretical FLOP advantage does not always translate to practical speedup. In contrast, our approach uses stan-

dard ViT blocks without any custom modules or architectural "bells and whistles", ensuring compatibility with widely used hardware accelerators and making deployment straightforward. Our method achieves competitive accuracy while significantly reducing network depth. For instance, DeiT-6-R and D-MAE-6-R achieve 70.2% and 72.4% top-1 accuracy, respectively, using only 3.1M parameters and 595M FLOPs, outperforming RePa-DeiT-Tiny/0.5 (69.4%) while remaining competitive with EdgeViT-XXS (74.4%)—despite using a simpler, fully transformer-based design. Compared to the 12-layer baselines DeiT-Tiny (72.2%) and D-MAE-Tiny (78.4%), our 6/4/3-layer variants reduce parameter count and FLOPs by up to 66%, while maintaining similar accuracy.

Table 3: Transfer evaluation on classification tasks and dense-prediction tasks. Self-supervised pre-training approaches generally show inferior performance to the fully-supervised counterpart. Top-1 accuracy is reported for classification tasks and AP is reported for object detection (det.) and instance segmentation (seg.) tasks.

| Methods | Datasets | | | | | COCO | |
|---|---|---|---|---|---|---|---|
| | Flowers | Pets | CIFAR10 | CIFAR100 | iNat18 | (det.) | (seg.) |
| *supervised* | | | | | | | |
| DeiT-Tiny Wang et al. (2023) | 96.4 | 93.1 | 96.1 | 85.8 | 63.6 | 40.4 | 35.5 |
| D-DeiT-6-R | 94.8 | 87.5 | 95.4 | 81.3 | 62.5 | 38.1 | 33.5 |
| D-DeiT-4-R | 94.1 | 79.3 | 93.1 | 75.6 | 60.7 | 35.1 | 31.4 |
| D-DeiT-3-R | 91.5 | 77.1 | 90.4 | 70.3 | 54.5 | 31.1 | 27.1 |
| *self-supervised* | | | | | | | |
| D-MAE-Tiny Wang et al. (2023) | 95.2 | 89.1 | 95.9 | 85.0 | 63.6 | 42.3 | 37.4 |
| D-MAE-6-R | 94.2 | 82.8 | 95.2 | 81.8 | 62.8 | 38.0 | 33.6 |
| D-MAE-4-R | 93.1 | 76.4 | 93.8 | 76.6 | 61.9 | 35.3 | 31.8 |
| D-MAE-3-R | 90.2 | 75.1 | 90.4 | 68.0 | 53.2 | 29.2 | 26.7 |

**Transfer Learning**: We evaluate transfer performance on Flowers Nilsback & Zisserman (2008), Pets Parkhi et al. (2012), CIFAR-10 Krizhevsky et al. (2009), CIFAR-100, and iNat-18 Van Horn et al. (2018) (Table 3). Both D-DeiT-6-R and D-MAE-6-R remain competitive with their 12-layer baselines, while the 4- and 3-layer variants are underperformed on several datasets. For instance, D-MAE-3-R achieves only 75.1% on Pets. On COCO Lin et al. (2014) detection and segmentation with Mask-RCNN $1\times$ He et al. (2017) referring to configurations provided by Li et al. (2021a), D-MAE-6-R achieves 38.0 AP (det.) and 33.6 AP (seg.), close to the MAE-Tiny baseline (42.3 / 37.4). This demonstrates that re-parameterized backbones preserve the spatial information required for dense prediction.

Table 4: Effect of Joining Stage on ImageNet-1K fine-tuning (Top-1 %) for D-MAE-6-R. All experiments use a total fine-tuning of 300 epochs and 10k-step warmup 50k-step adjustment with a linear lambda scheduler. #epochs-to-70% indicates the number of fine-tuning epochs required to reach 70% Top-1 accuracy.

| Joining Stage | Top-1 (%) | #epochs-to-70% |
|---|---|---|
| After pre-training (A) | **72.4** | 260 |
| Fine-tune begin (B) | 71.5 | 265 |
| Mid fine-tune (C, inserted at the 150th epoch) | 70.1 | 295 |
| Final stage of fine-tuning (D) | 68.9 | - |

**Effect of Joining Stage**: We investigate how the stage at which the joining operation is introduced affects fine-tuning. Intuitively, applying joining too early may interfere with stable representation learning, whereas applying it too late may reduce the model's ability to adapt to downstream tasks. We consider four strategies: **A**, applying joining immediately after pre-training and maintaining it throughout fine-tuning, enabling progressive joining from the start and facilitating smooth task adaptation; **B**, introducing joining only at the beginning of fine-tuning, so that the model is optimized jointly with the task objective from the outset; **C**, delaying joining until the middle of fine-tuning (150th epoch), under the assumption that joining is most effective once the model has largely transitioned to task-specific representations; and **D**, applying joining only at the final stage, thereby preserving maximum flexibility during fine-tuning and constraining the model only at the end. Our results in Table 4 indicate that introducing joining immediately after pre-training achieves the most consistent gains, suggesting that the re-parameterization mechanism benefits from guiding the optimization trajectory early on, while overly delaying the joining reduces its effectiveness.

**Lambda Scheduling and Diversity Regularization**: We investigate the effect of different lambda scheduling strategies and the addition of diversity regularization on the effectiveness of re-parameterization. To reduce redundancy among branches during and after joining, we introduce a diversity regularization that penalizes squared cosine similarity between branch features. Given $n$ feature representations $\{f_1, f_2, \ldots, f_n\}$ from different branches, we compute the average pairwise cosine similarity: $\mathcal{L}_{\text{div}} = \frac{1}{\binom{n}{2}} \sum_{i<j} \mathbb{E}\big[\cos(f_i, f_j)^2\big]$, averaged across attention and FFN layers and added to the training objective with weight $\alpha$ =0.05. This encourages complementary representations that can be better fused after joining.

Table 5 shows three main findings. First, instant joining severely degrades performance, confirming the necessity of a progressive schedule. Second, among schedulers, linear interpolation achieves the best overall accuracy and stability across model scales, while cosine, exponential, and square-root variants yield comparable but slightly weaker results. In particular, exponential and square-root schedules degrade most notably as the number of branches increases (e.g., D-MAE-3-R). Third, adding diversity regularization consistently improves results across all schedules on small-scale datasets such as CIFAR-10 (e.g., +0.6-1.6% for D-MAE-3-R), but brings

Table 5: Ablation on joining configurations (scheduling functions and implement of diversity regularization), using D-MAE-6-R, D-MAE-4-R and D-MAE-3-R pretrained via distillation on ImageNet-1K and evaluation on CIFAR10. Lambda scheduling implementation details are shown in Appendix B.2.

| Lambda sche. func. | Div. reg. | Accuracy Top-1 (%) | | |
| | | D-MAE-6-R | D-MAE-4-R | D-MAE-3-R |
|---|---|---|---|---|
| no joining ($\lambda = 0$) | ✗ | **95.9** | **94.0** | **92.2** |
| instant joining ($\lambda = 1$) | ✗ | 91.0 | 86.8 | 83.4 |
| linear | ✓ | 95.2 | 93.8 | 90.4 |
| | ✗ | 94.9 | 92.9 | 89.8 |
| cosine | ✓ | 93.9 | 92.2 | 90.2 |
| | ✗ | 93.9 | 91.7 | 89.7 |
| exponential | ✓ | 94.1 | 91.6 | 87.8 |
| | ✗ | 94.3 | 91.2 | 84.4 |
| sqrt | ✓ | 94.6 | 91.7 | 88.9 |
| | ✗ | 94.5 | 91.4 | 87.8 |

subtle benefit on larger-scale tasks (e.g., ImageNet-1K, COCO), as analyzed in Appendix C.4. Overall, linear scheduling combined with diversity regularization achieves the best balance between accuracy and stability, and is adopted as the default configuration in all experiments.

**Depth versus Branching**: We disentangle the contributions of depth and branching structures. Table 6 compares pure depth scaling, parallel branching without re-parameterization, and our proposed re-parameterized models. Simply increasing depth without branches yields limited improvements, while adding parallel branches without joining incurs large parameter and FLOP overheads. In contrast, our re-parameterization strategy consistently achieves higher accuracy with fewer parameters and significantly reduced computation. For example, D-MAE-6-R improves Top-1 accuracy by +3.4% over its depth-scaled baseline while using only half the compute compared to the naive multi-branch alternative. This demonstrates that re-parameterization not only enhances representational capacity but also provides a more efficient trade-off between accuracy and model complexity.

Table 6: Ablation study on ImageNet-1K. **Block-1**: Same-param Tiny backbone with $\times n$ parallel branches. **Block-2**: pure depth-scaling baselines. **Block-3**: our re-parameterized D-MAE-R. All models are distilled from the same MAE-Base teacher and fine-tuned for 300 epochs.

| Method | #Param. | FLOPs | Top-1 (%) |
|---|---|---|---|
| *Block-1* | | | |
| D-MAE-6-2 | 5.7M | 1.08G | 75.4 |
| D-MAE-4-3 | 5.7M | 1.08G | 71.4 |
| D-MAE-3-4 | 5.7M | 1.08G | 68.8 |
| *Block-2* | | | |
| D-MAE-6 | 3.1M | 556M | 68.6 |
| D-MAE-4 | 2.2M | 381M | 61.0 |
| D-MAE-3 | 1.7M | 293M | 53.8 |
| *Block-3* | | | |
| **D-MAE-6-R** | 3.1M | 595M | **72.4**(+3.8) |
| **D-MAE-4-R** | 2.2M | 415M | **67.0**(+6.0) |
| **D-MAE-3-R** | 1.7M | 322M | **60.5**(+6.7) |

## 5 CONCLUSIONS & DISCUSSIONS

In this work, we introduced a progressive re-parameterization strategy for ViT) that trains multiple parallel branches and gradually fuses them into a single streamlined model for deployment. By explicitly addressing the three core barriers to transformer re-parameterization—non-linear activations, LN, and MHSA concatenation—our method enables exact algebraic merging without approximation loss. This allows us to train wide, expressive multi-branch models and deploy ultra-shallow networks with dramatically reduced sequential depth, making ViTs far more suitable for real-time edge and mobile deployment.

Our ability to drastically reduce sequential depth aligns well with *analog and photonic computing platforms*, where parallelism and massively concurrent operations are natural advantages. Photonic accelerators, in particular, excel at implementing large matrix multiplications with extremely high throughput but are hindered by the need for deep, strictly sequential processing pipelines. By collapsing a deep ViT into a shallow counterpart, our approach could significantly reduce latency bottlenecks and facilitate seamless mapping to optical cores or mixed-signal arrays, paving the way for real-time, energy-efficient inference in next-generation hardware. Additionally, extending our approach to multimodal and spatiotemporal transformers could yield benefits for domains such as video understanding, robotics, and large-scale edge analytics.

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

# A EXPERIMENT SETTINGS

## A.1 PRE-TRAINING

For a fair comparison with prior work, we strictly adhere to the training protocols specified in the corresponding baseline papers. For supervised learning, we adopt the common configurations outlined in Touvron et al. (2021). Our setup for MAE closely follows the original design in He et al. (2022), including the optimizer, learning rate schedule, batch size, and data augmentation strategies. To better accommodate smaller-scale ViT encoders, we incorporate several adjustments provided by Wang et al. (2023), aiming to improve training efficiency on lightweight models. Specifically, we employ an MAE decoder consisting of a single Transformer block with width 192, and set the masking ratio to 0.75, which we found to yield the best performance in our setting.

Table 7: Pre-training settings for DeiT.

| config | value |
| --- | --- |
| optimizer | AdamW |
| learning rate | 1e-3 |
| weight decay | 0.05 |
| batch size | 1024 |
| learning rate schedule | cosine decay Loshchilov et al. (2017) |
| warmup epochs | 5 |
| training epochs | 300 |
| momentum coefficient | 0.9 |

Table 8: Pre-training settings for MAE.

| config | value |
| --- | --- |
| optimizer | AdamW |
| base learning rate | 1.5e-4(batch size=256) |
| weight decay | 0.05 |
| optimizer momentum | $\beta_1, \beta_2 = 0.9, 0.95$ |
| batch size | 1024 |
| learning rate schedule | cosine decay |
| warmup epochs | 40 |
| training epochs | 400 |
| momentum coefficient | 0.9 |
| masking ratio | 0.75 |
| teacher model | MAE-Base |

## A.2 PROGRESSIVE JOINING

Since joining occurs immediately after the completion of pre-training, the training configuration largely follows that of the pre-training stage. Unless otherwise specified, we set the lambda warmup period to 10k steps, followed by an adjustment phase of 50k steps, and a linear lambda scheduler is adopted as the default across all experiments.

## A.3 EVALUATION

For self-supervised model fine-tuning, we follow the improved protocol proposed in Wang et al. (2023) for MAE models. The default hyperparameter settings are summarized in Table 9. We adopt the linear learning rate scaling rule, defined as $lr = \text{base } lr \times \frac{\text{batch size}}{256}$ Goyal et al. (2017), and apply layer-wise learning rate decay following Bao et al. (2021). In contrast, supervised models are evaluated directly without additional fine-tuning.

## A.4 TRANSFER LEARNING

For transfer learning experiments, we follow the standard setup on small-scale classification datasets (e.g., CIFAR-10/100, Flowers, Pets, Aircraft, Cars) and the protocol described in Wang et al. (2023). Each dataset is trained using its recommended recipe, with the learning rate and number of epochs adjusted proportionally to the dataset size. Specifically, we sweep over three candidate learning rates $\{1e-2, 3e-2, 1e-1\}$ and three candidate epoch settings $\{(50, 10), (100, 20), (150, 30)\}$ (total epochs, warmup epochs) for each dataset, while applying layer-wise learning rate decay factors of $\{1.0, 0.75\}$. Fine-tuning is performed using SGD with momentum 0.9 and a batch size of 512. We adopt a cosine learning rate schedule with linear warm-up, and all images are resized to $224 \times 224$. Standard augmentations, including random resized cropping and horizontal flipping, are applied, without additional regularization. This setup ensures that performance comparisons across tasks primarily reflect the effect of re-parameterization. Finally, iNat18 is evaluated under the same fine-tuning settings described above.

Table 9: Fine-tuning evaluation settings.

| config | value |
|---|---|
| optimizer | AdamW |
| base learning rate | 1e-3 |
| weight decay | 0.05 |
| optimizer momentum | $\beta_1, \beta_2 = 0.9, 0.999$ |
| layer-wise $lr$ decay | 0.85 |
| batch size | 1024 |
| learning rate schedule | cosine decay |
| warmup epochs | 5 |
| training epochs | {100, 300} |
| augmentation | RandAug(10, 0.5) Cubuk et al. (2020) |
| colorjitter | 0.3 |
| label smoothing | 0 |
| mixup Zhang et al. (2017) | 0.2 |
| cutmix Yun et al. (2019) | 0 |
| drop path Huang et al. (2016) | 0 |

For COCO object detection and instance segmentation we follow the 100-epoch fine-tuning recipe of Li et al. (2021a), but replace the backbone with our ImageNet-1K pretrained, joined architecture. To match the baseline setting of Wang et al. (2023), the all sides of the input images is resized to 640 pixels instead of 1024. Our implementation is built on detectron 2 Wu et al. (2019).

## B  JOINING STRATEGY

### B.1  ABLATION STUDY ON THE LENGTH OF PROGRESSIVE JOINING

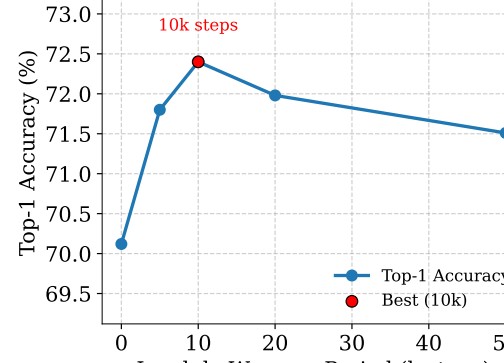

Figure 2: Effect of different Lambda Warmup Periods on ImageNet-1K D-MAE-6-R fine-tuning accuracy with linear lambda scheduler. The red point indicates the best performing warmup (10k steps).

We next investigate the duration and scheduling of the progressive joining process. For clarity, we define two separate periods: (1) the **Lambda Warmup Period**, which refers to the number of steps over which the branch joining coefficient $\lambda(t)$ gradually increases from 0 to 1. We evaluate six warmup durations: 0k (instant joining), 5k, 10k, 20k, and 50k steps. (2) the **Post-Joining Adjustment Phase**, during which $\lambda$ remains at 1 and the network can further fine-tune its parameters to optimize for the fully fused representation. In our experiments, we fix this phase to 50k steps. Results on ImageNet-1K (Table 2) show that a 10k-step Lambda Warmup Period achieves the best trade-off between training efficiency and final accuracy, and we adopt it as the default in all main experiments. Instant joining (0k warmup) destabilizes optimization and yields the worst accuracy, highlighting the importance of progressive joining. Shorter warmup periods (5k) mitigate instability

but remain suboptimal, while overly long warmups (20k, 50k) offer only marginal gains at the cost of unnecessarily extending the process.

## B.2 LAMBDA SCHEDULING FUNCTIONS

During the progressive joining process, we employ a scheduling function $\lambda(t)$ that controls how quickly multiple branches are fused into a single branch. Here, $t \in [0, 1]$ is the normalized training progress within the Lambda Warmup Period (e.g., 10k steps). After the warmup period, the model enters the Post-Joining Adjustment Phase, where $\lambda(t)$ remains fixed at 1 and the network continues to optimize with a fully fused representation.

We explore four different joining functions:

$$\text{Linear:} \quad \lambda_{\text{linear}}(t) = t \tag{10}$$

$$\text{Cosine:} \quad \lambda_{\text{cosine}}(t) = \tfrac{1}{2}\left(1 - \cos(\pi t)\right) \tag{11}$$

$$\text{Exponential:} \quad \lambda_{\text{exp}}(t) = 1 - e^{-5t} \tag{12}$$

$$\text{Square-root:} \quad \lambda_{\text{sqrt}}(t) = \sqrt{t} \tag{13}$$

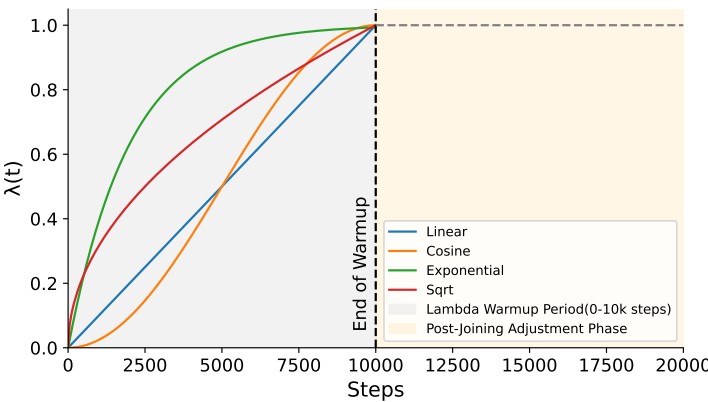

Figure 3: Visualization of different joining functions during the Lambda Warmup Period ($t \in [0, 10k]$). After the warmup (e.g., 10k steps), the Post-Joining Adjustment Phase begins with $\lambda = 1$.

Each of these functions provides a different trade-off between early and late joining: Linear lambda scheduler increases $\lambda$ at a constant rate. Cosine scheduler starts more smoothly and accelerates later, which can reduce early instability. Exponential scheduler fuses branches aggressively at the beginning and quickly approaches 1. Square-root scheduler emphasizes slower growth at the beginning but accelerates faster toward the end.

Figure 3 shows the behavior of these four joining functions over the Lambda Warmup Period, followed by the Post-Joining Adjustment Phase (where $\lambda = 1$). We observe that cosine and square-root schedules provide smoother transitions compared to the linear baseline, while exponential is more aggressive.

## C  LEARNED REPRESENTATION ANALYSIS

### C.1  FEATURE SPACE VISUALIZATION AND BRANCH DIVERSITY

To understand how progressive joining affects the representational capacity of our D-MAE architecture, we conduct a comprehensive analysis of branch representations at different training stages. We first examine the feature learning space of our pre-trained D-MAE-3-4 model (i.e., the base model before joining, which we refer to as D-MAE-3-R) through similarity visualization.

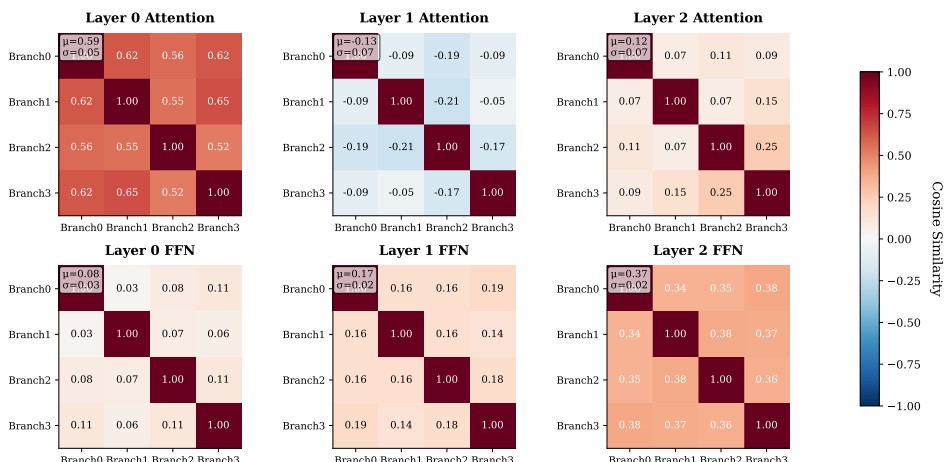

Figure 4: Feature similarity heatmap across branches in the pre-trained D-MAE-3-R model before progressive joining. The visualization shows cosine similarity between feature embeddings from attention and FFN outputs of each branch.

As shown in Figure 4, we compute cosine similarity between the attention and FFN outputs of each branch to generate this heatmap. Our analysis reveals a critical insight: while Layer 0 attention modules exhibit high inter-branch similarity (average = 0.59), which can be attributed to all branches receiving identical input tokens, other modules demonstrate relatively low similarity. This finding confirms that our base model's branches successfully learn distinct feature spaces before joining, enhancing the model's representational capacity through architectural diversity.

## C.2 WEIGHT SIMILARITY ANALYSIS: BEFORE AND AFTER JOINING

Since feature similarity analysis becomes less informative after joining (due to joined input sharing by weights making high similarity inevitable), we employ an alternative approach: measuring similarity between branch weights themselves.

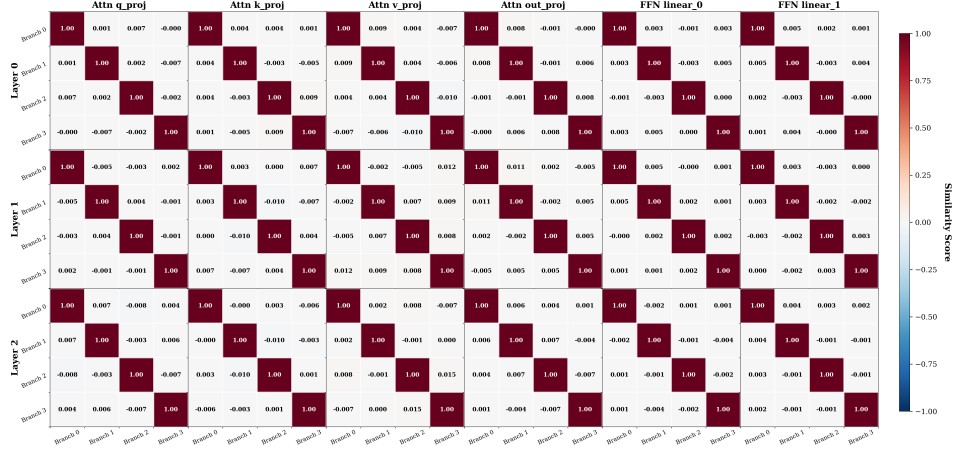

Figure 5: Weight similarity matrices across branches before progressive joining. This analysis corroborates the feature diversity observed in Figure 4, confirming that branches maintain distinct parameter spaces in D-MAE-3-4.

Figure 5 corroborates our observations from Figure 4, providing additional evidence that the base 3-layer, 4×-wide ViT model with branches effectively enhances representational capacity through maintained diversity.

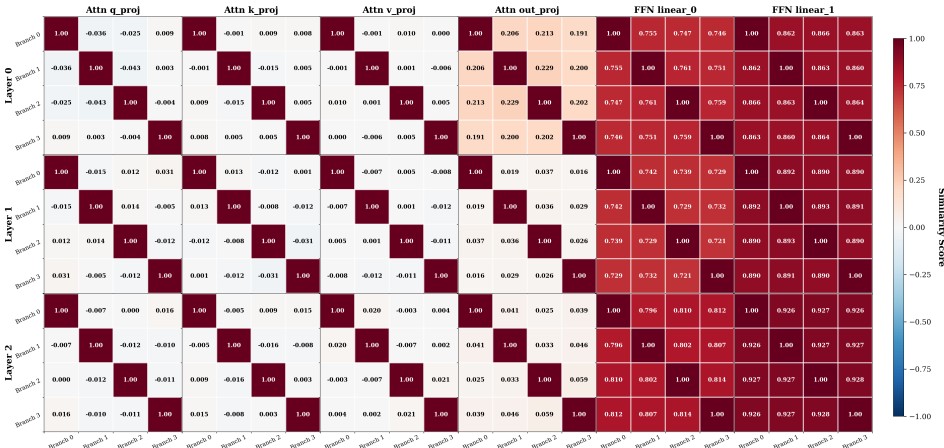

Figure 6: Weight similarity matrices after progressive joining (after Lambda Warmup Period of 10k steps and Post-Joining Adjustment Phase of 50k steps). Results show concerning overfitting patterns without diversity regularization, particularly in FFN modules.

### C.3    POST-JOINING OVERFITTING CHALLENGES

Figure 6 reveals a critical challenge in our progressive joining approach. After completing the Lambda Warmup Period (10k steps) and Post-Joining Adjustment Phase (50k steps) without diversity regularization, we observe alarmingly high similarity in FFN weights (0.725-0.928), suggesting potential overfitting behavior. Interestingly, our experiments reveal dataset-dependent recovery patterns.

**Large Dataset Recovery**: When continue fine-tuning on ImageNet-1K for 300 epochs without diversity regularization, FFN weight similarities naturally decrease to very low values ($< 0.02$), indicating that large-scale data provides sufficient regularization through its inherent diversity.

**Small Dataset Vulnerability**: On smaller datasets(e.g. CIFAR10 and CIFAR100) with transfer learning, FFN weight similarities remain critically high (0.98-1.0) when continue conduct fine-tuning, suggesting severe overfitting where branches collapse to nearly identical representations. This phenomenon likely occurs because limited data cannot provide the natural regularization effect observed with larger datasets.

### C.4    ANALYSIS AND DESIGN RATIONALE

The weight similarity patterns we observe can be understood through the lens of optimization dynamics. During progressive joining, the $\lambda$ parameter creates a continuous interpolation between independent branch optimization and shared parameter updates. Without explicit diversity constraints, the optimization naturally converges toward parameter alignment, as this minimizes the loss function while satisfying the joining objective. However, this convergence comes at the cost of representational diversity, which is particularly problematic when the training data lacks sufficient complexity to naturally encourage diverse feature learning. The dramatic difference between large and small dataset behaviors suggests that data diversity acts as an implicit regularizer, preventing catastrophic similarity collapse in parameter-rich scenarios. This analysis motivates our use of diversity regularization as an essential component of the progressive joining framework, ensuring that the computational benefits of joining do not come at the expense of representational capacity.

## D    MEMORY USAGE

We measure peak GPU memory consumption under identical settings on a single NVIDIA H100 GPU to evaluate the runtime cost of our re-parameterization. All measurements use an input image size of 224×224, a batch size of 256, and SGD (momentum = 0.9) for the training runs. No mixed-precision (AMP) was used; reported numbers are for FP32 to provide conservative (worst-case)

estimates. Peak memory for inference was measured in evaluation mode with `torch.no_grad()`; peak memory for training was measured during a forward+backward step. In both cases we reset and read `torch.cuda.max_memory_allocated()` to obtain the peak value (reported in GB).

The results (Table 10 and 11) show a modest training overhead for re-parameterized multi-branch models: the baseline D-MAE-Tiny uses 13.16 GB, while D-MAE-6-R and D-MAE-4-R use 14.43 GB ($\approx$+9.7%), and D-MAE-3-R uses 14.54 GB ($\approx$+10.5%). By contrast, inference peak memory is essentially unchanged: the baseline reports 1.30 GB and the re-parameterized variants report $\approx$1.29 GB ($\approx$0.99$\times$).

These findings indicate that the extra memory cost is concentrated in the training phase (due to additional activations and temporary buffers required by the multi-branch structure), whereas our progressive re-parameterization effectively folds branches for inference and therefore incurs negligible inference overhead.

# E ALGORITHMIC DESCRIPTION OF PROGRESSIVE REPARAMETERIZATION

The core training algorithm for our proposed progressive structural reparameterization is outlined in Algorithm 1. At a high level, the algorithm operates as follows:

Starting from the input, parallel transformer branches are initialized, and layer normalization is applied before branches split so that all branches receive identically normalized inputs. Within each layer of each branch, attention is computed in a blockwise manner, replacing the traditional concatenation of heads with independent projection and summation per head, as described earlier.

The main novelty arises in the attention computation: every head in every branch independently computes its own $Q$, $K$, and $V$ projections, then combines $Q$ and $K$ weights into a single matrix $W_{b,h}$ for that branch and head. This enables computation of attention scores as $XW_{b,h}X^T$, which are suitable for algebraic combination across branches. Progressive joining is achieved by mixing scores from both branches according to a schedule parameter $\lambda(t)$, with proper scaling to maintain numerically stable variance. These scores are softmaxed and the resulting attention is applied to values $V_{b,h}$, with output projections performed in a blockwise (head-wise) fashion.

The process also applies to the feedforward (FFN) sublayers, where branch merging occurs at the non-linear activation. Over the course of training, $\lambda(t)$ is smoothly increased so that, at convergence, all branches are merged and the final step algebraically collapses the model into a single-branch network by summing branch weights.

This procedure guarantees identical sequential and parallel computation at deployment while enabling richer representations during training. The pseudocode thus expresses: (1) blockwise attention and output projection, (2) branch-wise attention score mixing, (3) correct numerical scaling for score joining, and (4) final algebraic fusion for efficient inference. The entire routine is efficient and hardware-friendly, requiring no special kernels or additional barriers for parallel deployment.

Table 10: Peak GPU memory consumption (in GB) during training.

| Model | Params (M) | Peak Memory (GB) | Relative Overhead |
|---|---|---|---|
| D-MAE-Tiny | 1.08G | 13.16 | 1$\times$ |
| D-MAE-6-R | 3.1 M | 14.43 | 1.09$\times$ |
| D-MAE-4-R | 2.2 M | 14.43 | 1.09$\times$ |
| D-MAE-3-R | 1.7 M | 14.54 | 1.10$\times$ |

Table 11: Peak GPU memory consumption (in GB) during inference.

| Model | Params (M) | Peak Memory (GB) | Relative Overhead |
|---|---|---|---|
| D-MAE-Tiny | 1.08G | 1.30 | 1$\times$ |
| D-MAE-6-R | 3.1 M | 1.29 | 0.99$\times$ |
| D-MAE-4-R | 2.2 M | 1.29 | 0.99$\times$ |
| D-MAE-3-R | 1.7 M | 1.29 | 0.99$\times$ |

---

**Algorithm 1** Progressive Structural Reparameterization for Non-Deep Vision Transformers

---

**Require:** Input sequence $X \in \mathbb{R}^{N \times d}$, number of layers $L$, number of branches $B$, progressive parameter $\lambda(t)$

1: Initialize parallel transformer branches $\{\mathcal{B}_1, \ldots, \mathcal{B}_B\}$
2: **for** each training step $t$ **do**
3:     **for** each layer $\ell = 1$ to $L$ **do**
4:         **Layer Normalization:** Apply LN before branching, absorb $\gamma, \beta$ into adjacent linear layers
5:         **for** each branch $b = 1$ to $B$ **do**
6:             **Multi-Head Attention:**
7:             **for** each head $h = 1$ to $H$ **do**
8:                 Compute $Q_{b,h} = XW_{b,h}^Q$, $K_{b,h} = XW_{b,h}^K$, $V_{b,h} = XW_{b,h}^V$
9:                 Compute $W_{b,h} = W_{b,h}^Q (W_{b,h}^K)^T$
10:              Compute attention scores: $A_{b,h} = XW_{b,h}X^T$
11:              Progressive joining: $A_{b,h}^{\text{join}} = X(W_{b,h} + \lambda(t)W_{b',h})X^T$ for $b' \neq b$
12:              Scale scores: $A_{b,h}^{\text{join}} \leftarrow \frac{A_{b,h}^{\text{join}}}{\sqrt{1+\lambda^2(t)}\sqrt{d_k}}$
13:              Apply softmax: $S_{b,h} = \text{softmax}(A_{b,h}^{\text{join}})$
14:              Compute head output: $H_{b,h} = S_{b,h}V_{b,h}$
15:             **end for**
16:             Blockwise output projection: $O_b = \sum_{h=1}^H H_{b,h}W_{b,h}^O$
17:         **end for**
18:         **Feedforward Network:** Apply FFN with absorbed LN parameters
19:         **Progressive joining:** Merge branch outputs at non-linear activations (GELU, softmax) using $\lambda(t) = \frac{1}{2}\left(1 - \cos\left(\frac{\pi t}{T}\right)\right)$, where $t$ is linearly increased from 0 to T during training.
20:     **end for**
21: **end for**
22: **Collapse branches:** At inference, set $\lambda=1$ and combine all branch weights: $W^{\text{final}} = \sum_b W_b$
23: Deploy single-branch, non-deep transformer for efficient inference

---

