# OpenReview forum: "Non-Deep Vision Transformers"
_ICLR.cc/2026/Conference — ICLR 2026 Conference Withdrawn Submission_

### Official Review · Reviewer_QUPi · 2025-10-27

**Soundness:** 2
**Presentation:** 2
**Contribution:** 2
**Rating:** 2
**Confidence:** 5

**Summary:**

This paper proposes a progressive structural reparameterization framework that enables ViTs to operate at dramatically reduced depth  while maintaining accuracy comparable to standard 12-layer DeiT-Tiny models. The approach trains multi-branch parallel transformer blocks and then progressively merges then during training.  These branches are completely collapsed into single-branch layers for inference. Though the idea for reparameterization of multiple branches is not new, the idea of the paper is interesting in the sense that it adopts this idea for ViT architecture. The experimental results are competitive, though only tiny models are studied, resulting in limited impact. The paper also suffers from a few issues with presentation and references.

**Strengths:**

- Multi-branch parallel transformer blocks and with progressively merging is an interesting idea
- The experimental results are competitive, at least at Deit-tiny-level model sizes
- The method seems to be conceptually straightforward

**Weaknesses:**

-  The presentation needs some work, as some of the key technical information is missing. For example, Sec 3.2 does really show how LN is replaced by reparameterizable steps mathematically. Like how are sigma and mu in the original LN computed or treated in the new process? How is the mathematical equivalence achieved?
- The experiments are only conducted on tiny-level ViT architectures. I understand that the authors are upfront about this and do not try to claim wide applicability beyond such super lightweight models, however this is a main limiting factor of the method that restrict the scope of use and the scale of impact for the proposed method.
- Related to the previous one - bear in mind that methods like RePaViT also has a scaling factor which enables it to achieve increasing efficiency gains as the backbone architecture scales up, meaning that the comparison in this paper isn't necessarily providing a full picture of relative performance as it compares its strongest with other's weakest.
- The paper shows some sign of using LLM generation at least for its literature review. For example, Ln 503 "Ao Chang, Peng Zhao, Yingjie Pan, Xiaodong Tang, and Xingjun Gao. Repavit: Scalable vision transformer acceleration via structural reparameterization on feedforward network layers. arXiv preprint arXiv:2505.21847, 2024." is clearly a hallucinated version of the real paper in Ln 708 "Xuwei Xu, Yang Li, Yudong Chen, Jiajun Liu, and Sen Wang. Repavit: Scalable vision transformer acceleration via structural reparameterization on feedforward network layers. arXiv preprint arXiv:2505.21847, 2025."

**Questions:**

- How do you compute the sigma and mu? how is the algebraic equivalence maintained?
- How doe the model work for larger models?

---

### Official Review · Reviewer_3W9k · 2025-10-28

**Soundness:** 3
**Presentation:** 3
**Contribution:** 3
**Rating:** 4
**Confidence:** 5

**Summary:**

This paper introduces a novel structural reparameterization-based method that reduces the depth of ViT architecture while maintaining performance. Specifically, the proposed approach leverages multiple parallel branches in each ViT layer, and reparameterizes them into a single branch during testing/inference. With self-supervised learning technique, it achieves competitive performance as DeiT-Tiny yet incorporates only half of the total layers.

**Strengths:**

1. The proposed train-time multi-branch architecture with test-time reparameterization for efficient ViTs is novel and interesting.

2. The performance on DeiT-Tiny is strong.

**Weaknesses:**

1. __Unclear scalability:__ This work conducts experiments on DeiT-Tiny as the backbone only, thereby its effectiveness on standard-size ViTs (e.g., DeiT-Small, DeiT-Base) and large-size ViTs (ViT-L, ViT-H) remains unclear. Moreover, its performance on hierarchical Transformers (e.g., SwinTransformer) is also unknown. I suggest that the authors at least conduct experiments on larger ViTs under the same training settings to validate the scalability of the proposed method on really deep ViTs.

2. __Potential overclaim:__ It is worth noting that DeiT-Tiny is not universally regarded as a "deep ViT", and its performance cannot be considered as "strong". Therefore, the current empirical results cannot convince me of the claim that _your "findings challenge the conventional wisdom that transformer depth is essential for strong performance"_. Additionally, in contrast to CNNs where a deeper (>100 layers) network is usually more preferred, studies on ViTs commonly widen the architecture (e.g., the number of channels raises from 192 to 768 from DeiT-Tiny to DeiT-Base with the same number of layers at 12) rather than deepening it.

3. __Potential training overhead:__ During the training stage, the proposed method requires multiple parallel branches (two branches in the example, but could be more according to the description) in each MHSA and MLP layer, which can significantly increase the training cost.

4. __Lacking comparison to block pruning methods:__ The comparisons with prior works are poorly structured. Table 2 compares with efficient ViT architectures and reparameterization methods in the same place. I suggest the authors make three independent comparisons to:

   4.1. __Efficient ViTs__, such as EfficientViT, EfficientFormer, and models in Table 2, which demonstrates the state-of-the-art trade-off between model performance and efficiency realized by this work;

   4.2. __Reparameterization-based ViTs__, such as SLAB, RePaViT, and FastViT, which shows the superiority of the reparameterization strategy in this work and provides a new insight on leveraging reparameterization for ViTs;

   4.3. __Network pruning / dynamic depth / early stopping methods__, such as AdaViT, which contributes to understand how network depth affects the performance of ViTs

**Questions:**

1. In Equation 2, the LN parameters for MHSA layers are absorbed into the previous FFN layer's output projection. However, this can result in mathematically unequivalent output due to the shortcut in the FFN layer. Authors should explain how to deal with the shortcut when reparameterizing LN parameters.

---

### Official Review · Reviewer_zsiC · 2025-10-29

**Soundness:** 3
**Presentation:** 2
**Contribution:** 3
**Rating:** 2
**Confidence:** 5

**Summary:**

This paper proposes a novel reparameterization technique for ViTs. Specifically, it adapts the multi-branch design and test-time structural reparameterization concept from RepVGG to standard ViTs. First, it introduces two architectural modifications: (1) reparameterizing LayerNorm (LN) parameters into adjacent linear projections, and (2) enabling a hardware-friendly formulation of multi-head self-attention calculation. Second, it presents a progressive weight joining mechanism that gradually merges weights initialized in different branches, and eventually results in identical reparameterized weights among all branches. The over-parameterized network during training can be condensed into an efficient architecture during inference. Notably, when applied to DeiT-Tiny, the proposed approach achieves slightly higher accuracy with only half the number of layers. This work claims that a deep ViT can be effectively converted into a shallower counterpart with negligible performance degradation.

**Strengths:**

Overall, this work is innovative and interesting, as structural reparameterization has been rarely explored in the study of ViTs. The proposed mechanism offers a new perspective demonstrating that over-parameterization during training can also enhance performance in Transformer architectures. However, I remain doubting whether the proposed approach can be rigorously categorized as a true reparameterization method (see weaknesses below).

**Weaknesses:**

While this work is interesting, there are several significant problems/questions:

---

1. From my perspective, structural reparameterization typically refers to a technique where multiple branches with distinct weights (i.e., diverse learned knowledge) are trained, and subsequently merged into a single branch during inference. Such train-time over-parameterization enhances the backbone’s feature extraction capability. However, in this work, all branches have already converged to identical weights after training (i.e., when $\lambda=1$), which essentially learns the same representation across branches. Also, there is no weight merging occurring at inference; all merging operations (e.g., for $W^Q$ and $W^T$ , or $W_A$ and $W_B$) are performed either before training or during training. As a result, the benefit of multi-branch learning is unclear, and it is questionable whether this approach can be rigorously categorized as a structural reparameterization method.

   In addition, although it may not be directly relevant, the proposed architecture shares certain similarities with the mixture-of-experts (MoE) framework, where multiple branches are jointly trained. It is well known that MoE models require mechanisms to prevent branches from collapsing into identical behaviors, typically through expert diversification losses [1,2]. This principle appears contrary to the training objective of the current work, which encourages all branches to converge to identical weights.

---

2. In addition to W1, I am also interested in how the proposed reparameterization and architectural modifications are implemented in practice. I inspected the source code provided in the supplementary material but was unable to locate the components corresponding to these mechanisms, especially the reparameterization part. The available code appears to include only the multi-branch ViT implementation. This raises concerns about the reproducibility of the proposed method. Please correct me if I miss something.

---

3. The experimental scope is also quite limited, as the evaluation is conducted solely on the DeiT-Tiny backbone. Given that the proposed method functions more as a plug-in applicable to existing ViTs (where the number of parallel branches depends on the number of layers of the original and reparameterized models), validating it only on such a small-scale model is insufficient. It remains uncertain whether the approach generalizes effectively to larger and more applicable ViT architectures. My humble observation in the real-world projects is that no one is really gonna use DeiT-Tiny. You need to demonstrate the effectiveness of your approach on larger ViTs or hierarchical ViTs to signify its practical value.

   Furthermore, in terms of the efficiency-performance trade-off for a standalone architecture, DeiT-6-R still lags behind FastViT [3], which also incorporates structural reparameterization and is tailored for edge deployment. Specifically, FastViT-T8 achieves 75.6% accuracy with 3.6M parameters and 0.7G FLOPs, whereas DeiT-6-R attains only 70.2% accuracy with 3.1M parameters and 0.6G FLOPs. Considering the higher training cost of DeiT-6-R compared to FastViT-T8, the benefit brought by this work becomes marginal.

---

4. A minor presentation issue lies in the layout of Table 1, which is somewhat confusing at first glance. In Table 1, it is unclear whether the models listed are compared against the DeiT-Tiny backbone (12 layers) or only within each block of rows. Additionally, the grey-shaded rows appear visually separated, though they should logically belong to the corresponding groups with the same number of layers.Otherwise, the use of bold fonts for all grey-shaded rows seems inconsistent and potentially misleading.

---

References:

[1] Rajbhandari, Samyam, et al. "Deepspeed-moe: Advancing mixture-of-experts inference and training to power next-generation ai scale." ICML, 2022.

[2] Dai, Damai, et al. "Deepseekmoe: Towards ultimate expert specialization in mixture-of-experts language models." arxiv, 2024.

[3] Vasu, Pavan Kumar Anasosalu, et al. "Fastvit: A fast hybrid vision transformer using structural reparameterization." ICCV, 2023.

**Questions:**

Please refer to the weaknesses above. Additionally, I am a bit wondering how lambda changes would affect the performance. In appendix B.2 Figure 3, there is only the visualization of lambda warmups with no model performance corresponding to each one. The figure itself does not really illustrate anything useful.

I am open to reconsider my rating if concrete responses to my concerns.

---

### Official Review · Reviewer_2YPv · 2025-11-01

**Soundness:** 2
**Presentation:** 2
**Contribution:** 2
**Rating:** 4
**Confidence:** 4

**Summary:**

This paper studies the problem of reducing the depth of vision transformers while maintaining the performance. To be specific, the authors propose a structural reparameterization approach that enables training of parallel-branch transformer architectures which can be collapsed into efficient single-branch networks during inference. It allows  the reparameterization of both attention and FFN modules. Empirical results on ImageNet-1k validates the efficacy of the proposed method.

**Strengths:**

1. The direction of reducing the depth of vision transformers seems interesting and indeed worths exploration. The depth of deep networks plays a more essential role compared to width in terms of practical efficiency and it would bring more benefit if the depth of deep networks can be reduced during inference without information loss.
2. It is appreciated that the authors have conduct experiments on different tasks, including ImageNet classification, fine-grained classification, semantic segmentation and object detection.
3. The writing is clear and the paper is easy to follow. The overall structure is well organized and the idea is presented in a coherent manner.

**Weaknesses:**

1. The results in Tab. 1 is unconvincing. The authors finetune and distill from D-MAE and compare with DeiT-Tiny in Tab. 1 which may be misleading as they have very different backbones and trained with different learning objectives. One can tell from Tab. 2 that D-MAE full model has the accuracy of 78.4% and the proposed D-MAE-6-R has the performance of 72.4% which leads to 6% accuracy decrease on ImageNet-1K. If we consider the effect of knowledge distillation, the gap would be larger. In other words, the performance of the proposed method may not be satisfactory enough.
2. There are limited model compression baselines shown in the paper. The proposed method can be regarded as a model compression method and a practical question is: does the proposed method has advantages over existing model compression techniques, like pruning, distillation and quantization?
3. The scaling behavior of the proposed method is underexplored. The authors have only validated their performance on ViT-Tiny level model and the experiments on larger scale models are missing.
4. There is additional parameters and overhead introduced compared to depth-scaling baselines. It can be observed from Tab. 1 that the proposed method results in additional parameters and overhead compared to depth-scaling baselines and it is better to provide some explanations on this part.

**Questions:**

1. The reviewer wonders what is the training details of the depth-scaling baselines without re-parameterization? Is it fine-tune from pre-trained model and has it leveraged knowledge distillation in training as well?

---

### Comment · Area_Chair_MK5q · 2025-11-23
**Reviewer-Author Discussion**

Hi Reviewers,

Please kinly and actively participate in the review-author dicussion, raise your further concerns so that the authors can explain more, and make your final decisions.

---

### Note · Authors · 2025-11-23

I have read and agree with the venue's withdrawal policy on behalf of myself and my co-authors.